# Vojta Therapy Affects Trunk Control and Postural Sway in Children with Central Hypotonia: A Randomized Controlled Trial

**DOI:** 10.3390/children9101470

**Published:** 2022-09-26

**Authors:** Sun-Young Ha, Yun-Hee Sung

**Affiliations:** 1Department of Physical Therapy, Graduate School, Kyungnam University, Changwon 51767, Korea; 2Department of Physical Therapy, College of Health Sciences, Kyungnam University, Changwon 51767, Korea

**Keywords:** hypotonia, postural sway, abdominal muscle, trunk control, gross motor function

## Abstract

(1) Background: Decreased trunk stability is accompanied by delay in motor development in children with central hypotonia. We investigated the effect of Vojta therapy on trunk control in the sitting position in children with central hypotonia. (2) Methods: In 20 children with central hypotonia, Vojta therapy was applied to the experimental group (n = 10) and general physical therapy to the control group (n = 10). The intervention was applied for 30 min per session, three times a week, for a total of six weeks. We assessed abdominal muscle thickness, trunk control (segmental assessment of trunk control), trunk angle and trunk sway in a sitting position, and gross motor function measure-88. (3) Results: In the experimental group, the thicknesses of internal oblique and transversus abdominis were significantly increased (*p* < 0.05). The segmental assessment of trunk control score was significantly increased (*p* < 0.05), and the trunk sway significantly decreased (*p* < 0.05). Gross motor function measure-88 was significantly increased (*p* < 0.05). (4) Conclusions: Vojta therapy can be suggested as an effective intervention method for improving trunk control and gross motor function in children with central hypotonia.

## 1. Introduction

Hypotonia is a general term used to refer to decreased tone in the extremities, trunk, or craniofacial skeletal muscles. It can be discovered at birth or later in childhood [1]. Hypotonia is broadly divided into four categories: central nervous system, peripheral nerves, neuromuscular junction, and muscle [2,3]. Among them, hypotonia of central origin accounts for about 68–88% [3]. Central hypotonia is mainly seen in chromosomal abnormalities, genetic and neurological pathologies, and metabolic disorders [4]. It is characterized by trunk muscle weakness, loose ligaments, decreased endurance, and decreased muscle strength [5].

Trunk control is defined as the ability of the trunk muscles to erect the body and perform selective movements [6], and the sitting position achieved by controlling the trunk muscle is essential for daily activities [7]. Trunk control in normal developing infants occurs sequentially, one segment at a time, until they sit independently [8]. In addition, trunk control is closely related to the gross motor function [9,10]. In children with central hypotonia, the spine is erected by creating a frog-leg posture through hip abduction and hip external rotation [11]. As compensation for straightening the trunk, the trunk is leaned forward while supporting the floor with the arm, and a round shoulder appears [11,12]. Therefore, the development of the trunk muscles is an important factor in improving motor function in children with central hypotonia [13]. Among the trunk muscles, the abdominal muscle is reported to be an important determinant for effectively performing daily life and maintaining correct posture [14]. Ultrasonography was used in studies to determine the muscular structure of the abdominal muscle in children [15,16], because it has the advantages of low cost, non-invasive evaluation, and deep muscle measurement [17].

Cognitive task training, sensory integration therapy, neurodevelopmental treatment, gait training, and Vojta therapy are intervention methods applied to improve the motor function of children with central hypotonia [18,19,20,21]. Among them, Vojta therapy stimulates specific stimulation zones to induce normal motor development [22], suppresses the patient’s incorrect movement, and promotes correct posture control [23,24]. In addition, it affects the improvement of motor function through the activation of the trunk muscles and the deep spinal muscles; it also emphasizes the spinal segmental movement through the extension of the spine [16,25].

Trunk control is an important factor in motor development in children with central hypotonia, and the pattern of sitting, standing, and gait of the children may differ depending on the muscle tone. Nevertheless, most studies on trunk control in children have focused on children with cerebral palsy (CP). In addition, studies about Vojta therapy and its effect on the improvement of motor function in children with central hypotonia are few. Therefore, this study aimed to determine the effect of Vojta therapy on trunk control, trunk sway, and gross motor function in children with central hypotonia.

## 2. Materials and Methods

### 2.1. Participants

This study was conducted in children with central hypotonia who were admitted at M Hospital in Busan, Korea. Participants were assessed prior to intervention and randomly assigned to one of two groups. For the groups, numbers written on sealed paper were distributed by independent researchers; odd numbers applied Vojta therapy (experimental group), and even numbers applied general physical therapy (control group). The evaluator was not involved other than for the evaluation of this experiment, and the therapist was a person with more than 3 years of experience in pediatric physiotherapy working in a hospital. The inclusion criteria were as follows: (1) children with hypotonia who were diagnosed with developmental delay by a rehabilitation doctor before the age of 5 years; (2) children who can sit on their own or maintain a sitting position; and (3) children who can follow simple verbal instructions from the researcher. The exclusion criteria were as follows: (1) children with spastic CP; (2) children diagnosed with attention deficit hyperactivity disorder or autism; (3) children with vision or hearing problems; (4) children with acute fever or inflammation; (5) children with uncontrolled seizure; and (6) children with scoliosis.

This clinical study was designed as a randomized controlled trial, which was conducted for 6 weeks. The sample size was calculated using G*power software (G*power version 3.1.9.7, Heinrich-Heine-Universität Düsseldorf, Düsseldorf, Germany). The effective size was 1.212 according to the previous study by Ha and Sung [25]. The power and alpha levels were set at 0.80 and 0.05, respectively. Each group required at least 10 participants, with a total of 20 participants. Of the 21 participants, 1 was excluded due to vision problems. Twenty participants were assigned randomly to either the experimental group (n = 10) or the control group (n = 10). The intervention was applied in each group at 30 min per session, 2 sessions a day, 3 times a week, for 6 weeks. In both groups, participants’ abdominal muscle thickness, Segmental Assessment of Trunk Control (SATCo) score, trunk angle, trunk sway, and Gross Motor Function Measure-88 (GMFM-88) score were examined before and after the intervention. The procedure and purpose of the study were explained to the participants and their parents, who signed a consent form. This study was approved by the Research Ethics Committee of Kyungnam University (1040460-E-2022-003).

### 2.2. Outcome Measures

#### 2.2.1. Abdominal Muscle Thickness

Abdominal muscle thickness was evaluated by scanning images in B mode using the SONON application and a 10 MHz linear probe of ultrasonography (SONON, Healcerion, Seoul, Korea). The participants maintained the hooklying position for measurement at rest and lifted the head toward the knee according to the examiner’s instructions for measurement during the activity [26]. During this time, the thickness of the right abdominal muscle was measured, respectively. The thicknesses of the external oblique (EO), internal oblique (IO), and transversus abdominis (TrA) were measured by placing the probe horizontally on the upper part of the iliac crest from the center line of the right armpit (Figure 1). The thickness of the rectus abdominis (RA) muscle was measured by placing the probe vertically 2 cm next to the navel. For muscle thickness, a vertical line was drawn at the center of the image to connect the upper boundary end point of the fascia shown as a white image to the lower boundary end point. The thickness of all muscles was measured at the end of the exhalation, and to minimize bias, an operator with extensive experience in ultrasound techniques measured the probe pressure to a minimum. The measurement was performed using calipers built into the application, and the measured value was recorded in mm [16]. The change rate of abdominal muscle was calculated by the following [27]:Change rate of abdominal muscle (%) = (active thickness − rest thickness)/rest thickness × 100

#### 2.2.2. Segmental Assessment of Trunk Control (SATCo)

We evaluated the static, active, and reactive control of the trunk in seven segments using SATCo, respectively: (1) static control is the ability to maintain a neutral trunk posture vertically for 5 s; (2) active control is the ability to maintain a neutral trunk posture without compensation during movement from side to side of the head; and (3) reactive control is the ability to maintain and recover the trunk posture when pushing the xiphoid process and both acromion from the front, and pushing both shoulders to the left and right. The seven levels were as follows: head control (C7), upper thoracic control (T3), middle thoracic control (T7), lower thoracic control (T11), upper lumbar control (L3), lower lumbar control (S1), and maintenance without support. A total of 10 small circular stickers (2 cm × 2 cm × 2 cm) were attached in six segments (C7, T3, T7, T11, L3, S1) to the right ear tragus, right temporal lobe sulcus, right ASIS, and right greater trochanter of the femur to track the exact spot on the body. At the seven levels, if static, dynamic, and reactive controls were maintained, 1 point was provided for each, and if not, 0 points were provided; the total score was 20 points (reactive control of C7 was not tested). A higher score means better trunk control. The intra- and inter-rater reliabilities of SATCo were 0.98 and 0.84, respectively [28]. We recorded the video of the SATCo process using Galaxy Tab S3 (Samsung, Seoul, Korea) placed on the right side (distance, 3 m; height, 0.7 m) and behind (distance, 1 m; height, 0.5 m) each participant. The trunk angle and sway distance (sagittal and coronal planes) were measured using the video.

#### 2.2.3. Postural Sway in Sitting

Dartfish software program (Dartfish 7, Lausanne, Switzerland) was used to measure the sway distance through the recorded video during the reactive control of SATCo in the sagittal and coronal planes. The sway distance of the head in the sagittal plane was measured by the distance covered by the sticker attached to the right ear tragus as it moved back and forth. The sway distance of the trunk in the coronal plane was measured by the distance covered by the sticker attached to the 11th thoracic spine as it moved right and left [29].

The sway distance formula using the Dartfish software is as follows: √(X0 − X1)^2^ + √(Y0 − Y1)^2^, where X is the value of the X-axis over time and Y is the value of the Y-axis over time [30].

#### 2.2.4. Gross Motor Function

The GMFM-88 was used to evaluate gross motor function. Evaluation items were divided into five areas (A: lying and rolling, B: sitting, C: crawling and kneeling, D: standing, E: walking, running, and jumping). The validity, inter-rater reliability, test–retest reliability, and intra-rater reliability of GMFM-88 were 0.91 [31], 0.77, 0.88, and 0.68, respectively [32].

#### 2.2.5. Trunk Angle

Dartfish software program was used to measure the trunk angle through the recorded video during the static control of SATCo in a sagittal plane. The trunk angle was measured by the angle between the line connecting the right ear tragus and the right greater trochanter and the horizontal line passing through the right greater trochanter [33].

### 2.3. Intervention

In the experimental group, Vojta therapy was performed by a therapist with more than 3 years of experience who attended the professional course of Vojta therapy, and all procedures were performed the same regardless of the order. Vojta therapy was divided into reflex turning 1 phase, reflex turning 2 phase, reflex creeping, and first position. The reflex turning 1 phase stimulated the breast zone in the supine position, and the reflex turning 2 phases stimulated the medial border, 1/3 of the scapula medial border, and the anterior superior iliac spine (ASIS) in the side-lying position. Reflex creeping stimulated the medial epicondyle of the humerus and calcaneus in the prone position. The first position was in the prone position on the table, with both knees bent and the upper limbs in the same position as reflex creeping. The stimulation zones were the ASIS and the medial epicondyle of the humerus. Depending on the posture, 10 min each for a total of 30 min were applied, and the reflex creeping or the first position was selected according to the characteristics of the children.

In the control group, general physical therapy was performed by a therapist with more than 3 years of experience in pediatric physical therapy. Trunk stabilization exercise, pelvic control exercise in a sitting position, lower extremity strengthening exercise, and balance exercise in a sitting position and standing position were applied for 30 min.

### 2.4. Statistical Analysis

IBM SPSS version 21.0 (IBM Corp., Armonk, NY, USA) was used for statistical analysis. The Shapiro–Wilk test was performed to test the normality of variables, and descriptive statistics were used for assessing the general characteristics of participants. The independent and paired *t*-tests were used to assess the difference between and within groups, respectively. All statistical significance levels (α) were set to 0.05.

## 3. Results

### 3.1. General and Clinical Characteristics of the Participants

The general and clinical characteristics of the participants are shown in Table 1.

### 3.2. Comparison of Change Rate of Abdominal Muscle Thicknesses

In the change rate of TrA and IO thicknesses, the experimental group was significantly thicker than the control group post-intervention (*p* < 0.05). In the control group, the IO and TrA thicknesses were significantly thinner after the intervention than before (*p* < 0.05). The changes (post–pre) in TrA thickness were significantly higher in the experimental group than in the control group (*p* < 0.05) (Table 2).

### 3.3. Comparison of SATCo Scores

The SATCo scores of the experimental and control groups were significantly increased within the group (*p* < 0.05) (Table 3).

### 3.4. Comparison of Trunk Angles in the Sagittal Plane during the Static Control of SATCo

At post-intervention, the experimental group had a significantly larger trunk angle than the control group in T3, T7, T11, L3, and S1 (*p* < 0.05). In the experimental group, the trunk angle of T3 was significantly increased within the group (*p* < 0.05). The changes (post–pre) in T3, T11, and L3 trunk angles were significantly larger in the experimental group than in the control group (*p* < 0.05) (Table 4).

### 3.5. Comparison of Postural Sway in the Sagittal and Coronal Planes during the Reactive Control of SATCo

In the sagittal plane for the experimental group, L3 and S1 were significantly decreased within the group (*p* < 0.05). The changes (post–pre) in L3 and S1 were significantly decreased in the experimental group compared to the control group (*p* < 0.05). In the coronal plane for the experimental group, L3 and S1 were significantly decreased within the group (*p* < 0.05). The change (post–pre) in S1 was significantly decreased in the experimental group when compared to the control group (*p* < 0.05) (Table 5).

### 3.6. Comparison of Gross Motor Function

The GMFM-88 and GMFM-sitting of the experimental and control groups were significantly increased within the groups (*p* < 0.05). The change (post–pre) in GMFM-88 score was greater in the experimental group than in the control group (*p* < 0.05) (Table 6).

## 4. Discussion

Efficient trunk control allows patients to perform various tasks in a vertical position without losing balance and plays an important role in motor development [34,35]. Children with central hypotonia have reduced trunk control and delayed motor development. In this study, we confirmed that Vojta therapy was effective with trunk control and postural sway by increasing trunk stability through contraction of the abdominal muscles.

The abdominal muscle controls and maintains spinal stability and balance. Children with central hypotonia show pelvic anterior tilt in the supine position due to abdominal muscle weakening. This posture extends the RA and TrA, which is necessary to keep the pelvis in a neutral position and reduce the reactivity of these muscles by inhibiting the stretch reflex [26,36]. In the study by Masaki et al. they found that the abdominal muscle thickness was lower in children with hypotonia than in typically developing children [37]; therefore, strengthening the abdominal muscle of children with central hypotonia could be an important treatment point. In the study by Ha and Sung, when the breast zone of reflex turning 1 phase was applied to healthy individuals, the thickness of TrA increased while the thickness of EO decreased; they reported that Vojta therapy induced selective abdominal muscle contractions such as abdominal hollowing [25]. Son et al. have reported that the dynamic neuromuscular stimulation applied to children with spastic CP increased the thickness of the TrA; they reported that breast zone stimulation would activate the deep trunk muscles, increasing intra-abdominal pressure and playing a major role in stabilizing the trunk [38]. Therefore, abdominal muscle is essential for improving trunk control, and through this, functional movements can be performed [7,39]. In the present study, when Vojta therapy was applied, the change rate of the thicknesses in the IO and TrA were increased. When the stimulation points of Vojta therapy are stimulated, deep IO and TrA are contracted through the diaphragm. This is thought to increase trunk control by acting like a corset that stabilizes the spine by increasing trunk pressure. Rather, the decrease in the change rate of the thicknesses in of IO and TrA in the control group is thought to be due to the absence of direct stimulation of deep muscle.

Trunk control is an essential component when performing activities in a sitting position [40]. When this is impaired, delays and limitations in motor development are observed that affect cognitive and emotional development [41]. Therefore, trunk control could be an important factor that determines motor function. Cutis et al. have found that improving a specific segment with limitations through the evaluation of each spine segment improves motor function and mobility [9]. In addition, an increase in postural control in one segment of SATCo improved the GMFM score by approximately 0.5–11 points; there is a correlation among children’s gross motor function, postural control, and vertical trunk control [9,10]. As a result of this study, the SATCo, trunk angle, and gross motor function were improved when applying Vojta therapy. This is thought to be the result of improved trunk control through increased abdominal muscle thickness. However, the SATCo score and gross motor function were improved even in the general physical therapy group. These results suggest that there was no difference between the groups, as the strengthening exercise applied in general physical therapy influenced trunk control and improved gross motor function.

Children with central hypotonia have difficulty in balance control because their dynamic control is reduced due to low muscle tone, and postural sway is greater in children with central hypotonia than typically developing children [42]. Therefore, it is important to reduce postural sway to improve balance control in children with central hypotonia [40], because postural control is directly related to trunk stabilization [43]. Watanabe et al. said that the slump sitting is a posture in which the pelvis is posteriorly tilted, and the upright sitting posture observed co-contraction of the abdominal muscles rather than the slump sitting, resulting in correct lumbar curvature [44]. Yoon et al. reported that when dynamic neuromuscular stabilization was applied to stroke patients, internal abdominal pressure increased, which improved the motor control of Tra/IO and diaphragm; that, in turn, affected the improvement of postural control [45]. The reduction in the sway distance in the spinal segment could be an improvement in the segment control [9]. The increase in the lumbar segment control affects posture [46,47]. Based on the results of this study, Vojta therapy group had a significantly decreased sway distance in the sagittal and coronal planes when supported by the L3 and S1 segments. The significant difference when supporting the lumbar spine is thought to be the result of increased trunk stability, as the transversus abdominis and internal oblique are anatomically connected to the thoracolumbar fascia through the lateral raphe [48]. Therefore, it is thought that the increase in abdominal muscle thickness affects the postural stability and sway distance of the vertebral segments.

The limitations of this study are that the number of participants is small, and the characteristics of the participants are different, so it is difficult to generalize the results of the study. In addition, it was difficult to control the variables that could affect the outcome in daily life other than the experimental intervention of the participants. Therefore, future studies should be conducted to overcome these limitations.

## 5. Conclusions

Applying Vojta therapy to children with central hypotonia increased abdominal muscle thickness and trunk control. We found that Vojta therapy influenced the improvement of body alignment, postural sway, and gross motor function by increasing trunk control. Therefore, Vojta therapy as an intervention for children with central hypotonia in clinical practice could have a considerable effect on improving motor function.

## Figures and Tables

**Figure 1 children-09-01470-f001:**
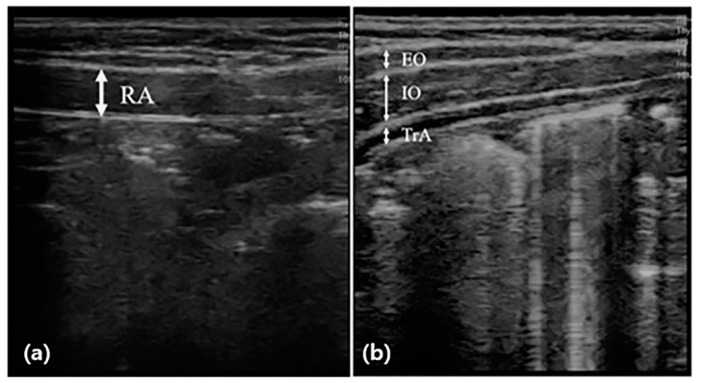
Measurement of abdominal muscle thickness through ultrasonography. The thickness of the rectus abdominal muscle (RA) was measured by placing the probe vertically (**a**), and the thicknesses of the external oblique (EO), internal oblique (IO), and transversus abdominis (TrA) were measured by placing the probe horizontally (**b**), and before and after the intervention.

**Table 1 children-09-01470-t001:** General and clinical characteristics of the participants (N = 20).

	Experimental (n = 10)	Control (n = 10)	*p*
Gender (M/F)	4 (40%)/6 (60%)	6 (60%)/4 (40%)	
Age (months)	45.00 ± 18.95	51.70 ± 27.42	0.533
Heights (cm)	91.89 ± 12.62	95.73 ± 16.10	0.560
Weight (kg)	13.21 ± 3.37	14.35 ± 5.87	0.601
Diagnosis			
Genetic disorder	2 (20%)	3 (30%)	
Charge syndrome	1 (10%)		
Angelman syndrome		1 (10%)	
Joubert syndrome	1 (10%)	1 (10%)	
Pierre Robin syndrome	1 (10%)		
Coffine–Lowry syndrome		1 (10%)	
Unknown	5 (50%)	4 (40%)	
Gross motor function			
Walk independently	3 (30%)	5 (50%)	
Walk-through walkers	5 (50%)	5 (50%)	
Maintain sitting position	2 (20%)		

**Table 2 children-09-01470-t002:** Comparison of change rate of abdominal muscle thicknesses (unit: %).

	Experimental			Control		
	Pre	Post	Post–Pre	Pre	Post	Post–Pre
EO	32.10 ± 35.77	36.83 ± 34.78	4.64 ± 31.28	30.92 ± 42.07	42.05 ± 45.55	11.13 ± 34.21
IO	32.21 ± 23.17	40.67 ± 30.88 ^†^	8.46 ± 39.70	26.57 ± 21.03	16.51 ± 24.64 *	−10.05 ± 17.55
TrA	27.04 ± 25.54	36.44 ± 28.28 ^†^	9.39 ± 44.25 ^†^	33.92 ± 22.77	2.85 ± 13.29 *	−31.07 ± 28.62
RA	31.33 ± 22.71	47.87 ± 45.13	16.54 ± 41.17	38.40 ± 23.33	36.11 ± 19.17	−2.29 ± 15.46

Values are presented as mean ± standard deviation, EO; external oblique, IO; internal oblique, TrA; transversus abdominis, RA; rectus abdominal muscle. * means significant difference within group (*p* < 0.05), ^†^ means significant difference from control (*p* < 0.05).

**Table 3 children-09-01470-t003:** Comparison of SATCo (unit: score).

Experimental			Control		
Pre	Post	Post–Pre	Pre	Post	Post–Pre
14.20 ± 3.57	15.10 ± 3.09 *	0.80 ± 0.76	15.70 ± 2.43	16.20 ± 2.14 *	0.50 ± 0.82

Values are presented as mean ± standard deviation, SATCo; segmental assessment of trunk control; * means significant difference within group (*p* < 0.05).

**Table 4 children-09-01470-t004:** Comparison of trunk angle (unit: °).

	Experimental			Control		
	Pre	Post	Post–Pre	Pre	Post	Post–Pre
T3	81.45 ± 7.44	83.90 ± 7.29 *^,^^†^	2.45 ± 4.36 ^†^	79.87 ± 5.56	78.42 ± 6.69	−1.45 ± 3.35
T7	83.72 ± 7.56	84.38 ± 5.52 ^†^	0.65 ± 5.18	80.51 ± 4.20	80.14 ± 6.03	−0.36 ± 5.59
T11	83.24 ± 9.19	85.73 ± 8.48 ^†^	2.49 ± 6.14 ^†^	81.65 ± 4.08	79.85 ± 6.04	−1.80 ± 5.52
L3	84.93 ± 7.47	87.11 ± 9.04 ^†^	2.17 ± 7.82 ^†^	85.71 ± 3.13	79.77 ± 5.74 *	−5.93 ± 5.95
S1	84.56 ± 5.73	85.35 ± 6.98 ^†^	0.78 ± 6.93	83.26 ± 6.69	79.97 ± 5.67	−3.28 ± 9.60

Values are presented as mean ± standard deviation, T; thoracic, L; lumbar, S; sacral bone; * means significant difference within the group (*p* < 0.05), ^†^ means significant difference from control (*p* < 0.05).

**Table 5 children-09-01470-t005:** Comparison of sway distance in sagittal and coronal plane (unit: cm).

		Experimental			Control		
		Pre	Post	Post–Pre	Pre	Post	Post–Pre
Sagittal plane	T3	25.57 ± 8.40 ^†^	21.46 ± 5.52	−3.0 ± 8.63	19.99 ± 5.22	18.55 ± 5.78	−1.44 ± 4.30
	T7	22.18 ± 6.56	18.35 ± 5.40	−3.82 ± 8.70	22.02 ± 8.53	21.94 ± 8.68	−0.07 ± 6.16
	T11	20.70 ± 5.51	20.57 ± 2.89	−0.13 ± 5.46	18.23 ± 3.54	19.01 ± 6.53	0.78 ± 7.53
	L3	24.41 ± 5.82	20.14 ± 3.56 *	−4.26 ± 6.61 ^†^	21.11 ± 5.36	21.70 ± 3.52	−0.59 ± 6.94
	S1	29.41 ± 3.16 ^†^	24.55 ± 3.36 *	−4.85 ± 4.46 ^†^	21.77 ± 6.61	22.70 ± 5.74	0.92 ± 3.45
Coronal plane	T3	10.82 ± 3.58	9.30 ± 2.43	−1.52 ± 4.04	11.00 ± 5.21	10.00 ± 2.71	−1.00 ± 4.05
	T7	12.08 ± 1.95	11.05 ± 3.12	−1.02 ± 4.29	11.42 ± 3.57	11.14 ± 1.91	−0.28 ± 2.74
	T11	12.85 ± 3.97	11.58 ± 3.28	−1.26 ± 5.84	11.22 ± 3.80	11.35 ± 3.94	0.13 ± 5.11
	L3	15.10 ± 6.95	11.87 ± 3.85 *	−3.22 ± 6.27	11.85 ± 4.43	11.28 ± 3.99	−0.57 ± 2.92
	S1	16.75 ± 4.40 ^†^	14.04 ± 4.40 *	−2.71 ± 4.69 ^†^	12.49 ± 4.56	13.85 ± 4.14	1.42 ± 5.03

Values are presented as mean ± standard deviation, T; thoracic, L; lumbar, S; sacral bone. * means significant difference within group (*p* < 0.05), ^†^ means significant difference from control (*p* < 0.05).

**Table 6 children-09-01470-t006:** Comparison of gross motor function (unit: %).

	Experimental			Control		
	Pre	Post	Post–Pre	Pre	Post	Post–Pre
GMFM-88	52.10 ± 19.12	54.46 ± 18.74 *	2.36 ± 1.07 ^†^	61.40 ± 17.12	63.12 ± 17.30 *	1.72 ± 0.49
GMFM-sitting	74.90 ± 25.88	78.32 ± 25.25 *	3.33 ± 2.53	84.74 ± 16.03	86.97 ± 14.40 *	2.22 ± 1.87

Values are presented as mean ± standard deviation, * means significant difference within group (*p* < 0.05), ^†^ means significant difference from control (*p* < 0.05).

## Data Availability

All relevant data are within the manuscript.

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
