# Peer review of "Vojta Therapy Affects Trunk Control and Postural Sway in Children with Central Hypotonia: A Randomized Controlled Trial"

_children, 2022, doi:10.3390/children9101470_

Round 1

Reviewer 1 Report

Thank you for the opportunity to review the article entitled "Vojta therapy effects trunk control and postural sway in children with central hypotonia: a randomized controlled trial." The aim of the study was to investigated the effect of the Vojta therapy on trunk control in sitting position in children with central hypotonia. The study was performed on 20 children  with central hypotonia. The authors assessed abdominal muscle thickness, trunk control (segmental assessment of trunk control), trunk angle and trunk sway in a sitting position,  and gross motor function measure. Based on the results obtained from the conducted studies, the authors conclude that the Vojta therapy can be suggested as an effective intervention method for improving trunk control and gross movement function in children with central hypotonia.

The article deals with an important and difficult to research topic of postural hypotonia in children. In general, the manuscript is written correctly, but several points need to be clarified and described in more detail.

The authors write about trunk muscles while examining only the abdominal muscles. There is a lack of information in the introduction and / or discussion where abdominal ultrasound has already been used in the pediatric population. It is also worth referring to the new data on the SWE of these muscles.It is worth noting that the assessment of the thickness itself may not provide complete information, as evidenced by the following tests:

https://pubmed.ncbi.nlm.nih.gov/33727639/

https://pubmed.ncbi.nlm.nih.gov/35701251/  - here, additionally, the risk of error due to the lack of automation of the procedurÄ™,

https://pubmed.ncbi.nlm.nih.gov/34208168/  - here as a limitation of the transducer pressure.

Abstract

Written correctly.

Introduction

Written correctly. Justifies taking up the research problem.

Materials and Methods

Line 64: Who were these people? Lack of information.

Line 65: A detailed description of randomization is needed.

Line 66: Who made the diagnosis of hypotonia and on what basis?

Line 75: Why just 20 participants? Is sample size specified?

Line 87: Only tested in isometric tension? Not tested at rest? The rest-isometric tension difference was not determined? Why?

Line 143: Too enigmatic description of the exercises. Who decided what exercises will be used? On what basis? Were the same procedures followed for all of them?

Discussion

Line 224: Why do the authors believe that strengthening the abdominal muscles in children with central hypotonia may be an important treatment point? The thickness of the abdominal muscles does not have to be related to their strength!

Line 268: „Therefore, it is thought that the increase in abdominal muscle thickness affected the 268 postural stability and sway distance of the vertebral segments.” This statement is not supported by any quotation. Why do the authors think so?

Author Response

Dear reviewer

We greatly appreciate your time and effort in reviewing this manuscript.

Based on the comments and suggestions received, we have made revisions and incorporated them in the revised manuscript.

Please find attached revised version of the manuscript with changes highlighted in red and point-by-point response to reviewer’s comments as below.  

We thank you again for your attention and would like to express our sincerest gratitude to the reviewer’s heartfelt comments.

Best regards,

Yun-Hee Sung

September 9, 2022

Reviewer 2 Report

This paper seems to be practically useful and interesting. However, there are some gaps which should be filled

1.The outcome measures section is missing

2. Scoring of applying tests are missing- the norms weren't presented; for reader is hard to decide if presented results means improvement or worsening

3. results: it will be easier for reader to see not only "was different" but if it was improved or not

Author Response

(The authors gave the same response as above.)

Round 2

Reviewer 1 Report

Thank you for being able to review the manuscript titled "Vojta therapy effects trunk control and postural sway in children with central hypotonia: a randomized controlled trial." The authors of this manuscript took into account all comments from the reviewer and provided full explanations of these comments. They also made appropriate manuscript corrections. Therefore, I consider the manuscript in its current form eligible for publication in the journal.